# Prevalence of Rotavirus-Associated Acute Gastroenteritis Cases in Early Childhood in Turkey: Meta-Analysis

**DOI:** 10.3390/children7100159

**Published:** 2020-10-02

**Authors:** Mustafa Güzel, Orhan Akpınar, Muhammet Burak Kılıç

**Affiliations:** 1Department of Medical Microbiology, Maltepe Medical Center, 34843 Istanbul, Turkey; 2Department of Microbiology, Health Sciences Institute, University of Süleyman Demirel, 32260 Isparta, Turkey; orhanakpnr@hotmail.com; 3Department of Business Administration, Faculty of Economics and Administrative Sciences, Quantitative Methods Unit, Mehmet Akif Ersoy University, 15100 Burdur, Turkey; mburak@mehmetakif.edu.tr

**Keywords:** acute gastroenteritis, children, early childhood, rotavirus infection, diarrhea

## Abstract

Background: Rotavirus is globally the most common viral pathogen in childhood gastroenteritis. This study aimed to estimate the number of Turkish children suffering from early-childhood gastroenteritis by rotavirus by performing a meta-analysis. Methods: Meta-analysis following the Preferred Reporting Items for Systematic Reviews and Meta-Analyses (PRISMA) guidelines was performed. Following the guidelines, primary studies were found reporting the prevalence of rotavirus gastroenteritis in Turkey. We performed a computerized search of published studies in national and international databases from 1990 to 2018. We selected 38 out of 721 studies for our study. Meta-analysis was carried out using R statistical software. The Cochrane *Q* statistic was calculated to assess the heterogeneity of the study results. Heterogeneity among studies was evaluated using the *I*^2^ statistic. Effect-size estimate was reported with 95% confidence interval. Results: On the basis of 38 selected articles, 80,113 children up to five years of age were diagnosed with symptoms of acute gastroenteritis, of whom the stool samples of 13,651 children were positive for rotavirus. The pooled prevalence of rotavirus was 19% in children younger than five years of age with acute gastroenteritis. In terms of seasonal prevalence, the highest prevalence rate was found in winter. Conclusion: This study supports the major prevalence of early-childhood gastroenteritis by rotavirus among Turkish children. Therefore, the decision to adopt immunization programs to prevent rotavirus infection might be helpful in Turkey.

## 1. Introduction

Diarrhea is one of the world’s top three infectious diseases that cause death. Viral gastroenteritis is most prevalent among infectious diarrhea, especially in childhood [1,2,3]. Diarrheal disease is the leading cause of death in early childhood, mostly in developing countries [1,3]. However, rotaviruses are the second leading cause of vaccine-preventable disease deaths after pneumococcal pneumonia in children under five years of age [4]. Rotavirus is highly contagious among children. Most children experience several episodes of rotavirus infection during the first years of life. Rotavirus is the leading etiologic agent of acute gastroenteritis in children under the age of five years. Due to rotavirus diarrhea, there are 111 million diarrhea attacks, 25 million outpatient clinics, and 2 million children hospitalized every year [3,5]. Diarrhea, vomiting, fever, abdominal pain, and dehydration occur in infants and children with rotavirus infection. In children under the age of two, the risk of dehydration is higher and may require hospitalization.

The virus is highly resistant to environmental conditions and can survive for months at room temperature [4]. Unlike other agents that cause gastroenteritis, rotaviruses are seen in the same frequency in developed and developing countries [1,3]. Rotavirus gastroenteritis causes mortality in developing countries where treatment options are insufficient and morbidity and economic losses in developed countries [1,5,6]. Although several studies were conducted in Turkey related to rotavirus infection, data are not available to develop preventive policies and programs. This study is the first meta-analysis on the prevalence of rotavirus gastroenteritis among the Turkish early-childhood population. This study’s primary purpose was to calculate pooled prevalence and evaluate the epidemiology of rotavirus infection in children younger than five years of age with acute gastroenteritis.

## 2. Methods

### 2.1. Systematic-Review Registration

A search protocol was prepared and registered with the PROSPERO database (PROSPERO/CRD42019117809. https://www.crd.york.ac.uk/prospero/export_details_pdf.php). Meta-analysis with the Preferred Reporting Items for Systematic Reviews and Meta-Analyses (PRISMA) guidelines was performed. Different phases of the study included searching for, screening, and selecting studies, and extracting, cleaning, and analyzing the data.

### 2.2. Search Strategy and Screening

We conducted a computerized review of the articles related to the subject published in national and international databases from 1990 to 2018. Ulakbim Turkish Medical Literature (ULAKBİM, Ankara, Turkey) and TürkMedline National Biomedical Periodicals (Turkish Medline) were used as national databases. As international databases, the literature was searched in the Embase, PubMed, Web of Science, Scopus, and Google Scholar databases. We used the following terms during the search: “rotavirus infections/epidemiology” OR “gastroenteritis/epidemiology” OR “diarrhea/epidemiology” AND “Turkey” AND “humans” AND “infant” OR “infant, newborn” OR “infant” OR “child, preschool”.

### 2.3. Study Selection

Prospective epidemiological, cross-sectional, and cohort studies that reported the prevalence of rotavirus infections in children with acute gastroenteritis were used. The titles and abstracts were scanned for possible inclusion. Meta-analysis was restricted to studies published in the English and Turkish languages. Studies on rotavirus epidemiology and prevalence only in Turkish children were evaluated.

### 2.4. Inclusion Criteria

The study participants had to be children under five years of age with acute gastroenteritis. positive and negative rotavirus results had to be reported. The rotavirus infection in these children was assessed by one of the acceptable laboratory diagnostic tests, including polymerase chain reaction (PCR), enzyme immunoassay (ELISA), immunochromatographic-method (IM) rapid test, and latex agglutination (LA). Articles written in English and Turkish with full texts were included.

### 2.5. Exclusion Criteria

Studies that had the following characteristics were excluded: studies using nonstandard methods, duplicate studies, studies published in languages other than English or Turkish, studies on ages other than early childhood, nonhuman studies, congress abstracts, review articles, meta-analyses, and articles only available in abstract form.

### 2.6. Quality Assessment

The researchers assessed the quality of articles with a checklist of Strengthening the Reporting of Observational Studies in Epidemiology (STROBE), using a PRISMA checklist and diagram. The majority of studies selected for this study had been assessed to be of moderate-to-good quality. Scores of 4 or less were evaluated as weak, between 5 and 7 as moderate, and between 8 and 10 as good. Articles thought to be of lower quality were excluded. The quality of the studies documented in the articles was primarily evaluated on the basis of tests used to detect rotavirus, age groups, and seasonal distribution.

### 2.7. Data Extraction and Definitions

Within the scope of the PRISMA report, exclusion criteria were determined at the first stage, and a literature review was carried out. In the next steps, the literature was re-examined with the exclusion criteria in mind; study data were collected, and statistical analysis was performed. Variables such as the name of the first author, duration of the study, year of study, the number of rotavirus gastroenteritis cases, and diagnostic methods were determined and recorded on the form. These data were obtained from only published articles in national and international journals. Both authors independently evaluated articles for compliance with the study criteria.

### 2.8. Meta-Analysis

We collected and cleaned the data on a Microsoft Excel sheet. We used the meta package of the R statistical software for standard meta-analysis [7]. Heterogeneity was assessed using the *Q* test and the *I*^2^ statistic. Effect-size (ES) estimate was reported with 95% confidence interval (CI). The *p*-value assessed the statistical heterogeneity between studies for *Q* test < 0.001 or *I*^2^ statistic of more than 50%. To assess any possible publication bias, the Begg rank-correlation and Egger weighted-regression methods, in combination with a funnel plot, were used. A *p* value of <0.05 was considered to be statistically significant.

## 3. Results

### 3.1. Description of Individual Studies

Overall, 2910 studies were identified with electronic searching. Out of these 2910 studies, 2189 were duplicates. In total, 721 potential articles were screened, and 650 were not included because they were not directly related to our study. After reviewing the full text of 71 related articles, they were checked for eligibility criteria. As a result, 19 studies were excluded. Lastly, 38 studies evaluated the epidemiology of rotavirus infection in children younger than five years of age with acute gastroenteritis. The study-selection process is shown in Figure 1, and the main characteristics of the selected studies are described in Table 1 [8,9,10,11,12,13,14,15,16,17,18,19,20,21,22,23,24,25,26,27,28,29,30,31,32,33,34,35,36,37,38,39,40,41,42,43,44,45].

### 3.2. Prevalence of Rotavirus Infection among Children under Five

On the basis of the 38 selected articles [8,9,10,11,12,13,14,15,16,17,18,19,20,21,22,23,24,25,26,27,28,29,30,31,32,33,34,35,36,37,38,39,40,41,42,43,44,45], 80,113 children under five years were diagnosed with symptoms of acute gastroenteritis, of whom the stool samples of 13,651 children were positive for rotavirus. The pooled prevalence of rotavirus was 18% (95% CI: 15–20) in children under five years with acute gastroenteritis (Figure 2). The prevalence of rotavirus was 12% (95% CI: 10–14) in children younger than two years (Figure 3). The prevalence of rotavirus was 5% (95% CI: 4–6) in children between two and five years with acute gastroenteritis (Figure 4).

### 3.3. Seasonal Distribution of Rotavirus Infection

In terms of seasonal prevalence, the highest prevalence rate was found in winter (4532/13,778; 32.9%). In addition, seasonal distribution was 3652/13,778 (26.5%) in spring, 3500/13,778 (25.4%) in autumn, and 2080/13,778 (15.1%) in summer. The highest incidence of rotavirus was observed in winter and spring. The ratio of rotavirus positivity was highest in December and January, while the lowest ratio was in July. This result was statistically significant (*p* = 0.001). The seasonal distribution is demonstrated in Figure 5.

### 3.4. Meta-Analysis Results

Heterogeneities between studies (Cochran’s *Q*-test, *I*^2^ = 99.0; *p* < 0.001) were found, so the random-effect model was used for meta-analysis. Details of the meta-analysis are summarized in Figure 2, Figure 3 and Figure 4. To assess possible publication bias, Begg rank correlation, Egger weighted regression tests, and a funnel plot were used. The results of Begg rank correlation were Z value for Kendall’s tau of 0.29 and *p* = 0.77 (*p* > 0.05), and those of the Egger weighted regression test were intercept −1.62, *t* = 0.40, and *p* = 0.69 (*p* > 0.05). No evidence of publication bias was observed. The funnel plot for the prevalence rate of rotavirus is shown in Figure 6.

## 4. Discussion

Acute gastroenteritis is the most critical health problem in early childhood. Diarrhea-related mortality was partially reduced due to advances in safe water and sanitation, as well as reduced undernutrition in early childhood [1,2]. Viral gastroenteritis is the most common cause of hospitalization in early childhood. Gastroenteritis caused by rotaviruses, also defined as a “democratic” virus, occurs at a similar frequency in developed and developing countries regardless of cleaning conditions [24]. Globally, most viral acute gastroenteritis cases are caused by rotavirus in children, especially in the first five years of life [1,2,6]. Rotavirus (RV) is one of the 13 gastroenteritis etiologic agents shown in the Global Burden of Disease Study [1,3,8]. Rotaviruses are the second leading cause of vaccine-preventable disease deaths after pneumococcal pneumonia in early childhood. In this context, the World Health Organization recommends that all countries add rotavirus vaccines to their national vaccination programs [4,46]. In Turkey, both monovalent and pentavalent vaccines were licensed in 2006. However, these vaccines are not funded by the Turkish national health system. The vaccine is not included in the national routine-vaccination program, but is available from pharmacies. Although not included in the routine national immunization program, the RV vaccination rate in Turkey was approximately 16% in 2018 [46]. Our study is the first and only meta-analysis of rotavirus-associated acute gastroenteritis cases in early childhood in Turkey.

According to data from the World Health Organization, the rotavirus-related mortality rate of children under five years was found to be 1.7 per 100,000 in 2013 in Turkey. Rotavirus-related deaths were reported to be 0.4% of all deaths of children under five years of age [1,3,8]. Turkey has a low child mortality and low adult mortality group in the European B group when analyzing countries by region and the type of death [47]. According to our study results, the overall proportion of rotavirus infection among gastroenteritis cases was 18% (95% CI: 15–20) in children under five years of age; in children under two years of age, this rate was 12% (95% CI: 10–14) These rates were found to be consistent with WHO data. Rotavirus gastroenteritis is very common in infants and young children. Early-childhood children, especially between the ages of six months and two years, are incredibly vulnerable. Due to the high incidence, morbidity, and mortality risk in this age group, research is focused on the pediatric-age group [3]. The rates of rotavirus-positive cases reported from other parts of the world are highly variable, varying between 20% and 50% in different countries and different pediatric-age groups [9]. This rate was reported as 27–52% in European countries [48]. In a study involving 20 countries in the Mediterranean region, the median rate of rotavirus diarrhea in children with acute gastroenteritis was 40% in hospitalized children and 23% in outpatients [49]. Another study covering the Middle East determined the overall percentage of rotavirus detection among gastroenteritis cases ranging from 16% to 61% [50]. A study of rotavirus infections and genotype distribution reported them to be 31.8% in children under five years of age and 25.9% in children under two years of age [51]. In our study, the seasonal distribution of rotavirus gastroenteritis was higher, especially in the spring and winter months. The proportion of children with gastroenteritis positive for the rotavirus antigen was found to be high between November and April, and it was highest in January and December. The lowest rates were recorded in July and August (Figure 5). The seasonal distribution of rotavirus gastroenteritis in European countries is similar to that in the United States [52]. Rotavirus gastroenteritis peaks in France and northern Europe in February or March. The Netherlands and Finland peak in March, while Spain generally has high rates in December and February. In Africa, the incidence of disease peaks during the dry season.

Interestingly, in December and January, the rate of rotavirus is the lowest in Iran, the eastern neighbor of Turkey [53,54]. As seen in the results of the study, although it is seen at the highest rates in winter and spring, the seasonality of this diarrhea form changes in different parts of the world. In our study, we determined a significant difference between disease rates in December, January, and February and its rates seen in other months.

### Study Limitations

Since the vaccination rates could not be reached in the articles, pre- and postvaccination evaluations could not be made.

## 5. Conclusions

Rotaviruses are the most critical form of viral gastroenteritis in childhood. They remain a significant public health concern because of their global social and economic burden. Accurately diagnosing and treating gastroenteritis factors in early childhood helps patients to quickly recover and avoid unnecessary costs. This study revealed that rotavirus is associated with gastroenteritis in Turkish children. We also demonstrated an epidemiologic picture of rotavirus gastroenteritis in Turkey. This study provides information about the different geographical areas of Turkey, and it is essential for drawing attention to the early-childhood route of viral gastroenteritis in Turkey.

## Figures and Tables

**Figure 1 children-07-00159-f001:**
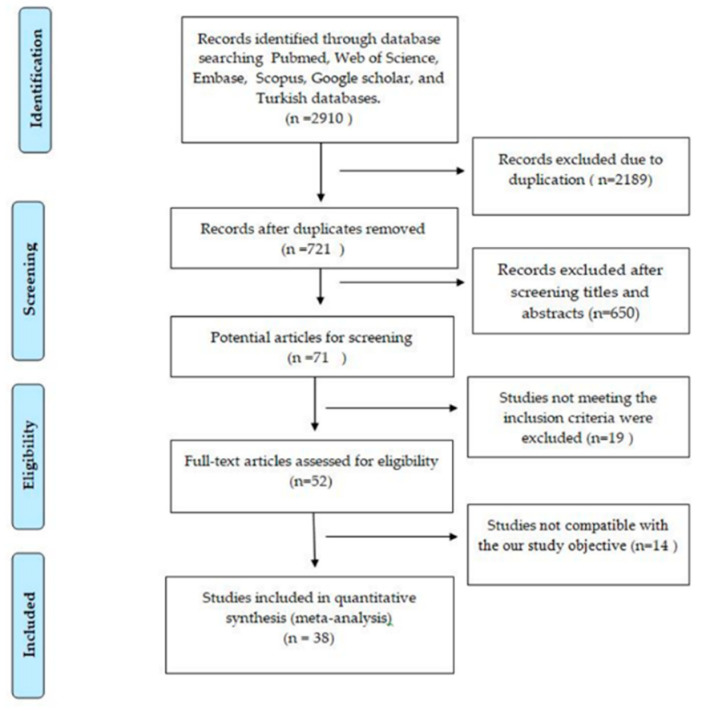
Flowchart of selection of utilized studies in meta-analysis.

**Figure 2 children-07-00159-f002:**
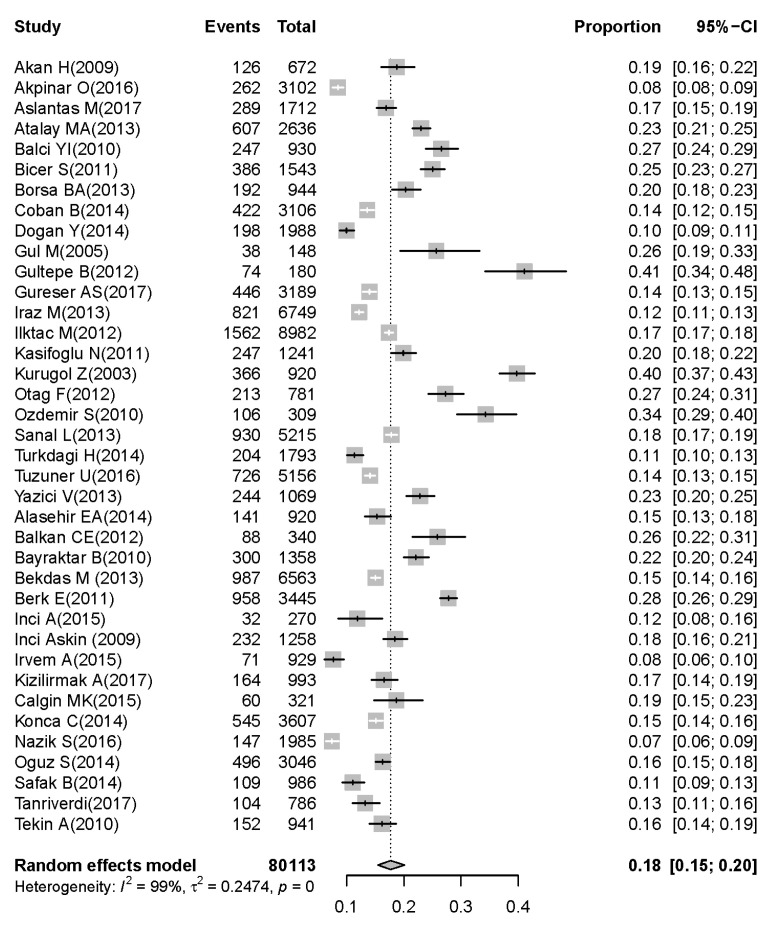
Prevalence of rotavirus-associated acute gastroenteritis cases in early childhood (0–5 years; results of meta-analysis for pooled prevalence with 95% confidence interval and its forest plot).

**Figure 3 children-07-00159-f003:**
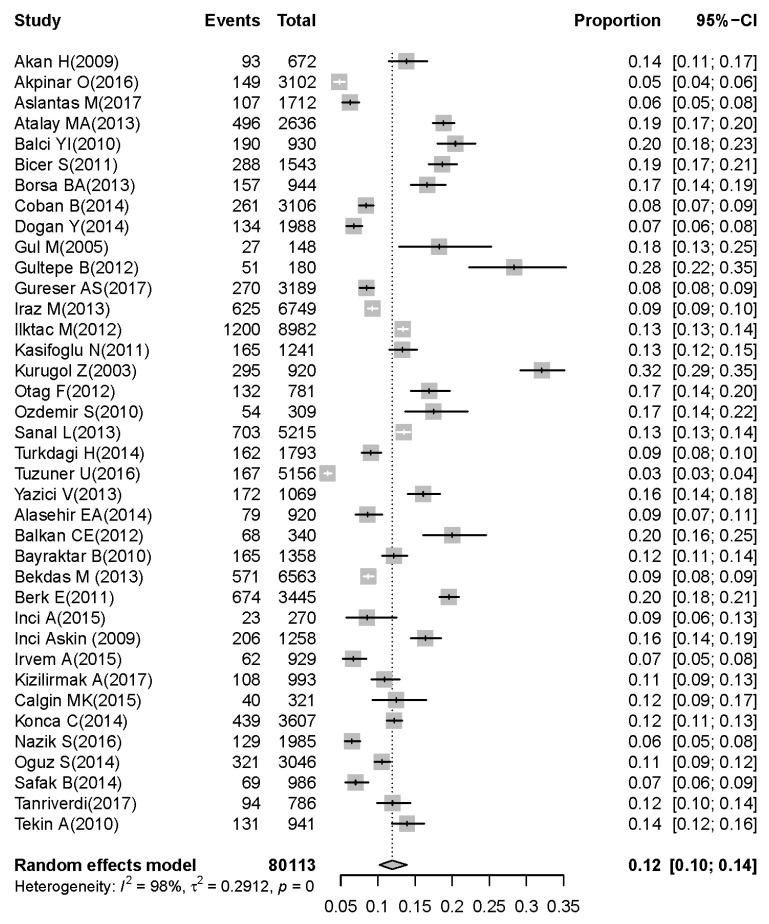
Prevalence of rotavirus-associated acute gastroenteritis cases in children aged 0–2 years.

**Figure 4 children-07-00159-f004:**
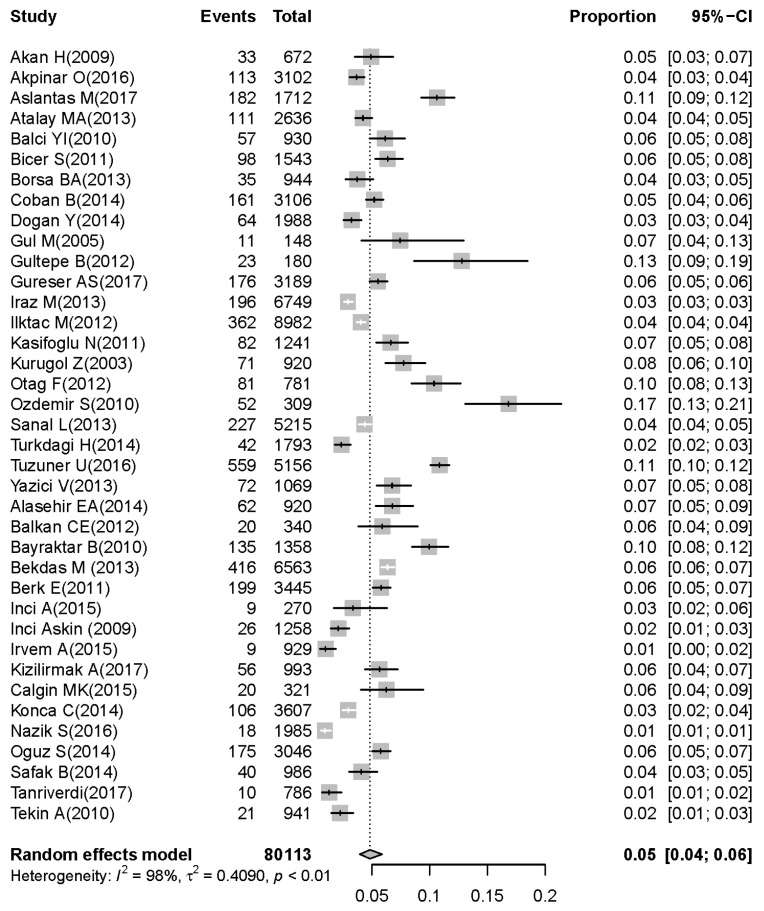
Prevalence of rotavirus-associated acute gastroenteritis cases in children aged 2–5 years.

**Figure 5 children-07-00159-f005:**
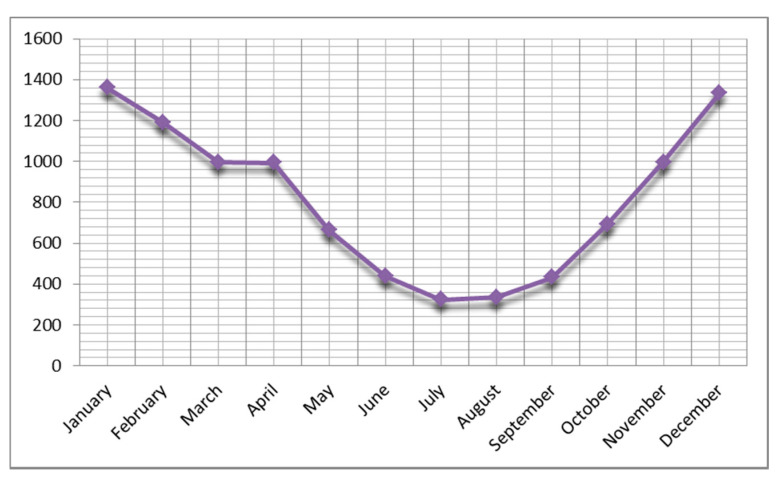
Distribution of prevalence of rotavirus gastroenteritis by month.

**Figure 6 children-07-00159-f006:**
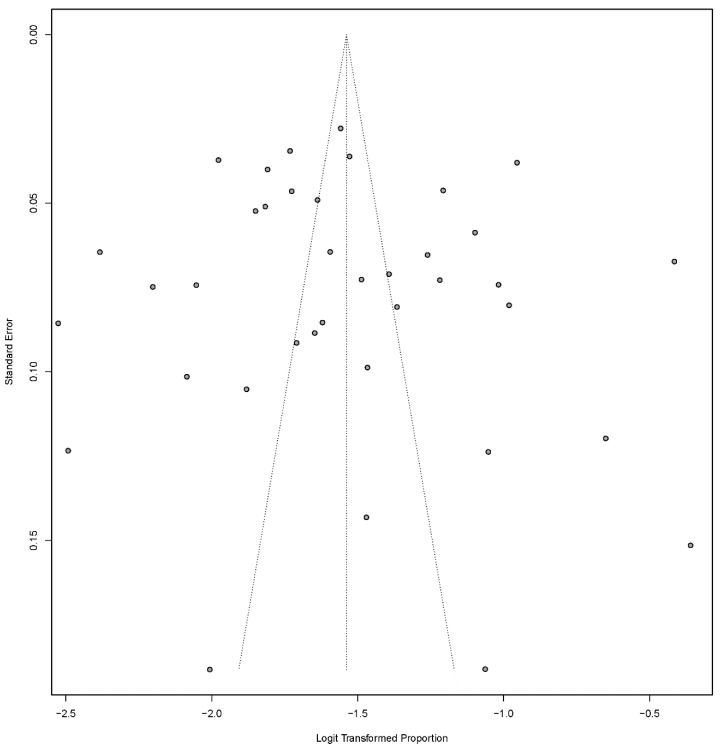
Funnel plot of prevalence of rotavirus-associated acute gastroenteritis cases in early childhood among Turkish children.

**Table 1 children-07-00159-t001:** Characteristics of studies included in meta-analysis.

No	First Author	Time of Study	Province	Total	Positive	Method	Ref. No
1	Akan H (2009)	2007–2008	İstanbul	672	126	IM	[8]
2	Akpınar O (2016)	2013–2014	Isparta	3102	262	IM	[9]
3	Aslantaş M (2017)	2013–2016	Düzce	1712	289	IM	[10]
4	Atalay MA (2013)	2009–2012	Kayseri	2636	607	IM	[11]
5	Balcı YI (2010)	2008–2009	Denizli	930	247	LA	[12]
6	Biçer S (2011)	2007–2008	İstanbul	1543	386	IM	[13]
7	Borsa BA (2013)	2010–2011	Mardin	944	192	IM	[14]
8	Çoban B (2014)	2008–2013	Antalya	3106	422	IM	[15]
9	Doğan Y (2014)	2012–2013	Gaziantep	1988	198	IM	[16]
10	Gül M (2005)	2003–2004	K.Maaraş	148	38	LA	[17]
11	Gültepe B (2012)	2009–2009	Van	180	74	LA	[18]
12	Güreser AS (2017)	2013–2014	Çorum	3189	446	IM	[19]
13	Iraz M (2013)	2011–2012	İstanbul	6749	821	IM	[20]
14	İlktaç M (2012)	2006–2010	İstanbul	8982	1562	IM	[21]
15	Kaşifoğlu N (2011)	2005–2011	Eskişehir	1241	247	ELISA	[22]
16	Kurugöl Z (2003)	2000–2000	İzmir	920	366	ELISA	[23]
17	Otağ F (2012)	2009–2011	Mersin	781	213	IM	[24]
18	Özdemir S (2010)	2008–2008	Mersin	309	106	ELISA	[25]
19	Şanal L (2013)	2011–2011	Ankara	5215	930	IM	[26]
20	Türkdağı H (2014)	2010–2013	Konya	1793	204	IM	[27]
21	Tüzüner U (2016)	2013–2015	Konya	5156	726	IM	[28]
22	Yazıcı V (2013)	2009–2011	Kocaeli	1069	244	IM	[29]
23	Alaşehir EA (2014)	2010–2013	İstanbul	920	141	IM	[30]
24	Balkan ÇE (2012)	2010–2011	Erzurum	340	88	IM	[31]
25	Bayraktar B (2010)	2008–2009	İstanbul	1358	300	IM	[32]
26	Bekdaş M (2013)	2007–2009	Bolu	6563	987	IM	[33]
27	Berk E (2011)	2009–2011	Kayseri	3445	958	IM	[34]
28	İnci A (2015)	2010–2013	Artvin	270	32	IM	[35]
29	İnci Aşkın (2009)	2007–2008	Konya	1258	232	IM	[36]
30	İrvem A (2015)	2012–2013	İstanbul	929	71	IM	[37]
31	KIızılırmak A (2017)	2014–2015	Düzce	993	164	IM	[38]
32	Çalgın MK (2015)	2013– 2015	Ordu	321	60	IM	[39]
33	Konca Ç (2014)	2012–2013	Adıyaman	3607	545	IM	[40]
34	Nazik S (2016)	2011–2015	Bingöl	1985	147	IM	[41]
35	Oğuz S (2014)	2011–2012	Ankara	3046	496	IM	[42]
36	Şafak B (2014)	2011–2013	Balıkesir	986	109	IM	[43]
37	Tanrıverdi (2017)	2014–2015	Samsun	786	104	IM	[44]
38	Tekin A (2010)	2008–2009	Mardin	941	152	IM	[45]

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
