# Peer review of "Prevalence of Rotavirus-Associated Acute Gastroenteritis Cases in Early Childhood in Turkey: Meta-Analysis"

_children, 2020, doi:10.3390/children7100159_

Round 1

Reviewer 1 Report

The authors have done extensive literature review on sero-prevalence of rotavirus associated acute gastroenteritis among early childhood in Turkey. Given that there is limited national level pooled estimate of rotavirus infection among children in the country, the result from this meta analysis could provide valuable information about the status of rotavirus infection in Turkey. However, there are some methodological issues that the authors need to address. 

  1. The authors did not clearly show what is known about the epidemiology of rotavirus infection and what information are lacking so that these study is intended to fill the existing knowledge/information gaps in the background of the study. Moreover, the authors need to show the current status of rotavirus immunization in the country as the vaccine was recommended to be given to under five children  globally by the world health organization (WHO) since 2009. 
  2. The authors clam that they followed PRISMA guideline to write this meta analysis study. However, Ethical considerations is not required in the PRISMA 2009 checklist for systematic review and meta analysis studies. On the other hand, Limitations of the study are not mentioned in the manuscript while it is recommended by the PRISMA guideline.  
  3. Authors have stated "nonstandard methods" in the exclusion criteria and authors did not include studies that involve molecular techniques such as PCR as a method of rotavirus detection. Is PCR a nonstandard method for rotavirus detection? Why authors are interested only in "sero-prevalence" instead of "prevalence"?
  4. Authors have used STROBE checklists for quality appraisal of individual studies. But, studies with low quality were not excluded according to the authors statement under the quality assessment section of the manuscript. So, why is that important to do the quality appraisal if decisions to either include or exclude are not made based on the quality assessment results? How can readers thrust the quality of the evidence in the presence of such type of selection/inclusion bias?
  5. The STROBE checklist is a qualitative checklist with 22 items. How did the authors set a cutoff for high quality, medium quality and low quality?
  6. The authors stated that "Two authors independently evaluated 
    the quality of the studies" under the data extraction section of the manuscript. However, this statement is better to be moved to the quality assessment section. In this same section, the authors stated that "For the controversial articles between the two authors, only the only reconciled were included in the study.". However, in case of disagreement between two independent reviewers, the recommendation is to look for another reviewer rather than excluding the article from the study as this may introduce selection bias. 
  7. The data extraction is not complete, it needs to include data by age (Under two, two-five and under five), vaccination introduction (before versus after), study setting (inpatient versus outpatient or Hospital based versus community based) etc. so that subgroup analysis can be considered during analysis. 
  8. The analysis methods employed does not sound correct. First of all the distribution of the data needs to be checked for heterogeneity. This can be achieved by qualitative approach (Galbraith plot) or quantitatively through forest plot coupled with the I2 test and Cochran’s Q-test. The authors did the later and that is good. However, in the meta analysis section of the manuscript authors stated as if Egger's and Begg's test are used for Heterogeneity test. 
  9. Authors needs to clearly show how publication bias was assessed using a funnel plot (qualitative method) substantiated with a quantitative method (Egger's or Begg's test) along with their respective interpretations. Authors can improve the funnel plot view by Logit transforming the standard error.
  10. Authors need to re-organize the result section of the report in to the following sub-titles: 3.) Results 3.1.) Description of the individual studies 3.2.) Prevalence of rotavirus infection among under five children 3.3.) Seasonal distribution of rotavirus infection other sub-titles can be included if the authors will consider doing more analysis (subgroup analysis, meta cumulative analysis and meta influence analysis) as per the review comments in the methods section. 
  11. Authors considered random effect model as sole remedy of heterogeneity. However, the source of the heterogeneity has to be addressed using meta-regression analysis also called subgroup analysis. Meta-cumulative and meta-influence analysis can be also done to show the trend over time and the influence of each study on the effect size respectively.   
  12. The number of articles in the Study selection flow diagram are not consistent. For example, records after duplicates removed were n=87 and then potential articles for screening were n=71. But, no reason was mentioned for exclusion of the 16 articles. The contribution of each database has to be also mentioned. 
  13. In the result section, authors need to include meta-regression, meta-cumulative and meta-influence results so that it can give a complete picture of the epidemiology of the disease in the country.
  14. Authors citation of the individual articles in the result section "[7-45]" seems incorrect. It is rather [9-47].
  15. The authors discussed something that is not their result. The discussion needs to be focused on the results of this study finding. Moreover, scientific biological or epidemiological explanations need to be given as a possible reason for disparities of the findings with other published data.

General 

  1. English grammar and typography needs further revision. It seems languages other than English language are used for example rotavirus appears to be "Rota virüs" in many sentences. In some cases rotavirus is miss-spelled as "Route virus". 
  2. All titles of figures (Figure 1-4) shall be at the bottom of the figure and legends need to be included for figures for ease understanding of the figures by the readers.

Author Response

Response to Reviewer 1 Comments

  1. The authors did not clearly show what is known about the epidemiology of rotavirus infection and what information are lacking so that these study is intended to fill the existing knowledge/information gaps in the background of the study. Moreover, the authors need to show the current status of rotavirus immunization in the country as the vaccine was recommended to be given to under five children  globally by the world health organization (WHO) since 2009.

Response 1: Information about the current status of rotavirus vaccination in the country has been added to the introduction and discussion section.

  1. The authors clam that they followed PRISMA guideline to write this meta analysis study. However, Ethical considerations is not required in the PRISMA 2009 checklist for systematic review and meta analysis studies. On the other hand, Limitations of the study are not mentioned in the manuscript while it is recommended by the PRISMA guideline. 

 Response 2: The part about the ethics committee has been removed. The limitations of the study have been added to the last part of the study.

  1. Authors have stated "nonstandard methods" in the exclusion criteria and authors did not include studies that involve molecular techniques such as PCR as a method of rotavirus detection. Is PCR a nonstandard method for rotavirus detection? Why authors are interested only in "sero-prevalence" instead of "prevalence"?

Response 3:PCR technique was not used in the publications we received in accordance with our criteria, so we could not include it. PCR test has been added to the ‘’Inclusion criteria’’section.

 The word seroprevalence was changed to prevalence.

  1. Authors have used STROBE checklists for quality appraisal of individual studies. But, studies with low quality were not excluded according to the authors statement under the quality assessment section of the manuscript. So, why is that important to do the quality appraisal if decisions to either include or exclude are not made based on the quality assessment results? How can readers thrust the quality of the evidence in the presence of such type of selection/inclusion bias?

Response 3:Sentence  that we wrote by mistake changed in Quality assessment section as follows. ‘’Articles thought to be of lower quality were excluded’’

  1. The STROBE checklist is a qualitative checklist with 22 items. How did the authors set a cutoff for high quality, medium quality and low quality?

Response 5: The scores in the results of the quality assessment evaluations made show the methodological quality of the studies, and the score ranges used in the study are as follows .Scores of 4 or less were evaluated as weak, between 5 and 7 as moderate, and between 8 and 10 asgood.

  1. The authors stated that "Two authors independently evaluated  the quality of the studies" under the data extraction section of the manuscript. However, this statement is better to be moved to the quality assessment section. In this same section, the authors stated that "For the controversial articles between the two authors, only the only reconciled were included in the study.". However, in case of disagreement between two independent reviewers, the recommendation is to look for another reviewer rather than excluding the article from the study as this may introduce selection bias.
  2. Response 6: Sentence"For the controversial articles between the two authors, only the only reconciled were included in the study " has been changed to ‘Both authors independently evaluated articles for compliance with the study criteria.’’.

  1. The data extraction is not complete, it needs to include data by age (Under two, two-five and under five), vaccination introduction (before versus after), study setting (inpatient versus outpatient or Hospital based versus community based) etc. so that subgroup analysis can be considered during analysis.

Response 7: Subgroup analysis considered .They were divided into subgroups as 0-2 age, 2-5 age and (0-5) age.

  1. The analysis methods employed does not sound correct. First of all the distribution of the data needs to be checked for heterogeneity. This can be achieved by qualitative approach (Galbraith plot) or quantitatively through forest plot coupled with the I2 test and Cochran’s Q-test. The authors did the later and that is good. However, in the meta analysis section of the manuscript authors stated as if Egger's and Begg's test are used for Heterogeneity test.

Response 8: Mentioned ambiguous statements have been clarified. Necessary data on heterogeneity are included in the analysis results.

  1. Authors needs to clearly show how publication bias was assessed using a funnel plot (qualitative method) substantiated with a quantitative method (Egger's or Begg's test) along with their respective interpretations. Authors can improve the funnel plot view by Logit transforming the standard error.

Improved funnel plot view by converting the logit standard error

  1. Authors need to re-organize the result section of the report in to the following sub-titles: 3.) Results 3.1.) Description of the individual studies 3.2.) Prevalence of rotavirus infection among under five children 3.3.) Seasonal distribution of rotavirus infection other sub-titles can be included if the authors will consider doing more analysis (subgroup analysis, meta cumulative analysis and meta influence analysis) as per the review comments in the methods section. 

Response 9: Sub-titles were rearranged as requested by the reviewer

  1. Authors considered random effect model as sole remedy of heterogeneity. However, the source of the heterogeneity has to be addressed using meta-regression analysis also called subgroup analysis. Meta-cumulative and meta-influence analysis can be also done to show the trend over time and the influence of each study on the effect size respectively. 

Response 11 : moderator analysis was made according to the publication year.

  1. The number of articles in the Study selection flow diagram are not consistent. For example, records after duplicates removed were n=87 and then potential articles for screening were n=71. But, no reason was mentioned for exclusion of the 16 articles. The contribution of each database has to be also mentioned.

 Response 12 : The numbers in the study selection flowchart have been corrected.

The flow chart has been rebuilt.

  1. In the result section, authors need to include meta-regression, meta-cumulative and meta-influence results so that it can give a complete picture of the epidemiology of the disease in the country.

 Response 13 : Sub-group analyzes were made and indicated in the conclusion part.

  1. Authors citation of the individual articles in the result section "[7-45]" seems incorrect. It is rather [9-47].

 Response 14 :Article citations corrected. The reference numbers in the table have been corrected.

  1. The authors discussed something that is not their result. The discussion needs to be focused on the results of this study finding. Moreover, scientific biological or epidemiological explanations need to be given as a possible reason for disparities of the findings with other published data

             Response 15: Discussion part was rearranged.

General

English grammar and typography needs further revision. It seems languages other than English language are used for example rotavirus appears to be "Rota virüs" in many sentences. In some cases rotavirus is miss-spelled as "Route virus".

Response: English grammar and typography has been revised. Errors in word rotavirus corrected

All titles of figures (Figure 1-4) shall be at the bottom of the figure and legends need to be included for figures for ease understanding of the figures by the readers.

Response: The titles of all figures are written at the bottom and explanations are included in the figures.

Reviewer 2 Report

The authors analyse in the manuscript the prevalence of rotavirus gastroenteritis in Turkey described in scientific literature during 1990 – 2018. The results should help in decision making concerning adoption of immunization programs against RVA.

The manuscript should be proofread by a native English speaker to correct some incomprehensible formulations.

According to the journal instructions for authors, the tables and figures should be named Table 1 or Figure 1 without parenthesis (“  “) and without abbreviations (“Fig-2” or “figure-3” is incorrect). Please, correct those throughout the manuscript. Also carefully revise the incorrect use of letter “ü” in the word “rotavirüs” or “figüre” (pages 1, 6, 7).

Carefully revise the use of word “rotavirus” in the whole manuscript. It should be written with a small letter except when at the beginning of the sentence. “Rota virus” or “rota virus” is also not correct.

Next I have some other comments which authors should address:

P1, L2: Title – I would recommend the authors to change the title of the manuscript as the word “Seroprevalence” is not suitable. Seroprevalence describes the concentration of some substance (most often of antibody) in serum. The authors, however, do not follow the seroprevalence of rotavirus (or antibodies against RVA) in serum but its presence in stool samples. More suitable words would be “Incidence”, “Presence” or just “Prevalence”. The same applies to other places in the manuscript, where “seroprevalence” is used (in the abstract; page 2, line 52; page 6, line 130, 133, 134; Figure 2

P1, L11-27: Abstract – Please, carefully revise the use of word rotavirus as instructed above. Next, correct the typing mistake in the word Turkish (“i” without a tittle). This mistake also appears on page 3 and in Figure 4. Please, rephrase the sentence on lines 23-24 “Seasonal prevalence among which the highest rate of prevalence was found in winter”. In this form it is not correct English.

P1, L31: Please, rephrase the sentence (e.g. Diarrhea is one of the world´s top three infectious diseases causing death… or similar).

P1, L35: Please, add citation.

P2, L45-47: “… rotaviruses are seen in the same frequency in developed and developing countries, regardless of socioeconomic conditions and hygiene measures [1,3].” Please, carefully revise this sentence as your own citations contradict this assertion:

Citation No. 1 (Methods): “The two leading risk factors for diarrhoea—childhood malnutrition and unsafe water, sanitation, and hygiene—were used…”

Citation No. 2 (Conclusions): “Reducing the diarrheal morbidity and mortality in low- and middle-income countries to the levels of that in high-income countries is within our grasp.”

The same comments concern P8, L157-158 (Discussion): Rotavirus indeed occurs at similar frequencies in most of the countries in the world, however, the cleaning, sanitation and socioeconomic conditions are of great importance and influence the seriousness of the disease as well as the mortality.

P2, L49: Please, rephrase the sentence.

P3, L106-107: Please, correct the sentence.

Figure 1: Correct the text in the right bottom frame: “month-by-case istribution, no data the distribution by age groups…”

Table 1: Please, correct the abbreviation of immunochromatographic method in the table – it should be IM not İM. Next, correct the numbers of references, which should be 7-45.

P7, L138: Please, rephrase the sentence.

P8, L167: Please, correct the mistake “Route virus-related” (Rotavirus-related).

P8, L175: Please, correct the typing mistake “… studies have focused on the pediatric age group.4”

P9, L181-184: “A recent study…” The cited study (Procop, 2001) is 19 years old, so it definitely is not recent.

References: Citation No. 3 and No. 5 mention the same publication. Please, omit one of them and correct the numbers accordingly in the manuscript.

Author Response

Response to Reviewer 2 Comments

1-The authors analyse in the manuscript the prevalence of rotavirus gastroenteritis in Turkey described in scientific literature during 1990 – 2018. The results should help in decision making concerning adoption of immunization programs against RVA.

Response 1: This study is the first meta-analysis of the prevalence of rotavirüs gastroenteritis among the Turkish early childhood population. This study's primary purpose is to calculate pooled prevalence and to evaluate the epidemiology of rotavirus infection in children younger than five years of age with acute gastroenteritis. Therefore, the decision to adopt immunization programs to prevent rotavirus infection might be helpful in Turkey.

2-The manuscript should be proofread by a native English speaker to correct some incomprehensible fo rmulations.

Response 2 English grammar and typography has been revised

3-According to the journal instructions for authors, the tables and figures should be named Table 1 or Figure 1 without parenthesis (“  “) and without abbreviations (“Fig-2” or “figure-3” is incorrect). Please, correct those throughout the manuscript. Also carefully revise the incorrect use of letter “ü” in the word “rotavirüs” or “figüre” (pages 1, 6, 7).

Response 3: Spelling errors were corrected by revising. Tables and figures were named without parenthesis and abbreviations.

4-Carefully revise the use of word “rotavirus” in the whole manuscript. It should be written with a small letter except when at the beginning of the sentence. “Rota virus” or “rota virus” is also not correct.

Response 4: The use of the word "rotavirus" in the whole article has been corrected.

Next I have some other comments which authors should address:

P1, L2: Title – I would recommend the authors to change the title of the manuscript as the word “Seroprevalence” is not suitable. Seroprevalence describes the concentration of some substance (most often of antibody) in serum. The authors, however, do not follow the seroprevalence of rotavirus (or antibodies against RVA) in serum but its presence in stool samples. More suitable words would be “Incidence”, “Presence” or just “Prevalence”. The same applies to other places in the manuscript, where “seroprevalence” is used (in the abstract; page 2, line 52; page 6, line 130, 133, 134; Figure 2

The word "Prevalence" was used instead of "Seroprevalence" in the entire article.

P1, L11-27: Abstract – Please, carefully revise the use of word rotavirus as instructed above. Next, correct the typing mistake in the word Turkish (“i” without a tittle). This mistake also appears on page 3 and in Figure 4. Please, rephrase the sentence on lines 23-24 “Seasonal prevalence among which the highest rate of prevalence was found in winter”. In this form it is not correct English.

Corected.

In terms of seasonal prevalence, the highest prevalence rate was found in winter.

P1, L31: Please, rephrase the sentence (e.g. Diarrhea is one of the world´s top three infectious diseases causing death… or similar).

Diarrhea is one of the world's top three infectious diseases that cause death.

P1, L35: Please, add citation.

Citation added

P2, L45-47: “… rotaviruses are seen in the same frequency in developed and developing countries, regardless of socioeconomic conditions and hygiene measures [1,3].” Please, carefully revise this sentence as your own citations contradict this assertion:

Citation No. 1 (Methods): “The two leading risk factors for diarrhoea—childhood malnutrition and unsafe water, sanitation, and hygiene—were used…”

Citation No. 2 (Conclusions): “Reducing the diarrheal morbidity and mortality in low- and middle-income countries to the levels of that in high-income countries is within our grasp.”

Corected as

It is unlike other agents that cause gastroenteritis; rotaviruses are seen in the same frequency in developed and developing countries. regardless of socioeconomic conditions and hygiene measures

The same comments concern P8, L157-158 (Discussion): Rotavirus indeed occurs at similar frequencies in most of the countries in the world, however, the cleaning, sanitation and socioeconomic conditions are of great importance and influence the seriousness of the disease as well as the mortality.

P2, L49: Please, rephrase the sentence.

Although various studies related to rotavirus infection, though made in Turkey. There are no data to develop preventive program policies. Although several studies conducted in Turkey related to rotavirus infection, the data are not available to develop preventive policies and programs

P3, L106-107: Please, correct the sentence.

 This data was obtained from only published articles in national and international journals. Two authors extracted data independently from all of the included studies. Two authors independently evaluated the quality of the studies. For the controversial articles between the two authors, only the only reconciled were included in the study. Articles were independently evaluated by both authors for compliance with the study criteria.

Figure 1: Correct the text in the right bottom frame: “month-by-case istribution, no data the distribution by age groups…”

İt is corected

Table 1: Please, correct the abbreviation of immunochromatographic method in the table – it should be IM not İM. Next, correct the numbers of references, which should be 7-45.

It is corected IM , the reference numbers in the table have been corrected [7-45]

P7, L138: Please, rephrase the sentence.

 Seasonal prevalence among which the highest prevalence was found in winter In terms of seasonal prevalence, the highest prevalence rate was found in winter

P8, L167: Please, correct the mistake “Route virus-related” (Rotavirus-related).

it is corrected

P8, L175: Please, correct the typing mistake “… studies have focused on the pediatric age group.4”

[4].

P9, L181-184: “A recent study…” The cited study (Procop, 2001) is 19 years old, so it definitely is not recent.

 A recent study of rotavirus infections and genotype distribution has been reported as 31.8% in children under five years of age and 25.9% in children under two years of age

References: Citation No. 3 and No. 5 mention the same publication. Please, omit one of them and correct the numbers accordingly in the manuscript.

it is corected

Round 2

Reviewer 1 Report

Though authors made some amendments to the original submission, there are still outstanding issues which needs authors attention. Below listed are further comments:

Abstract

Needs extensive English language edition

For example, page 1 line 17&18 “Of the remaining 152 articles, only 39 18 studies were included in this meta-analysis.” is not clear. Without mentioning any data prior to this sentence, authors began “of the remaining…….”. It needs edition. On the other hand, one of the statements, “The effect size was estimated by reported with its 95% confidence interval.”, it would rather be paraphrased as “Effect size estimate was reported with 95% confidence interval.”. In General, there are many typographic, contextual, and grammatical errors which needs extensive language edition.

Introduction

The statement “The 38 first infection tends to be the most violent because the body produces immunity against the virus.” Does not seem scientifically sound. Authors need to cite scientifically sound evidence that supports their statement or shall remove it.

Methods

In the inclusion criteria “The study period should include at least one year, the number of cases should be 100 and above, seasonal or month-by-case distribution, and the distribution by age groups.” Needs modification. For one thing, authors included cross-sectional studies, therefore the study may not last for a complete one year. On the other hand, the number of cases indicated “100” is not true. Authors have included several studies, 7 studies to be specific, with less than 100 cases. If authors intention is to specify about sample size, it has to be corrected accordingly.

Authors need to revise also “Quality assessment” section as per the above comment related to number of cases as quality assessment criteria.

Meta-analysis section, the statement “The effect size [ES] was reported to be 95% confidence interval [CI]” needs modification.

Results

The flow chart has lots of flaws. Authors mentioned they had 152 study articles in first paragraph of the result section, but it is 158 in the flow diagram. Moreover, I do not really think they had only this few number of articles in their first database search before removal of duplication. Usually the database gives large number of articles but through a step wise screening the number of study articles included will be small. Moreover, the statements in the flow diagram comment boxes are misspelled. Generally, the flow diagram needs critical revision.

Prevalence of rotavirus infection seems wrongly calculated. If we consider the overall prevalence in under five children to be 19%. There is no way that the prevalence in 0-2 YO =67% and 2-5= 25%. Because the overall prevalence would be >/= 25 according to the subgroup analysis data. Authors need to re-do the analysis.

Authors shall put figures and tables in brackets when they cite in the result section.

Figure 3 and Figure 4 have the same title. Authors need to check and correct.

Discussion

Authors have discussed their pooled rotavirus prevalence estimate with different reports and concluded that their prevalence is different from other study findings. For the differences authors mentioned “The reason for the different results from our study is the fiction and technique of the study” as a reason which is unclear and does not show how the study techniques were different.

Conclusion

There are still spelling errors such as “route virus” to mean rotavirus. Authors need to check the whole document for language edition preferably by English language expert.

Limitations of the study

This section shall precede the conclusion section. Moreover, the statements need rephrasing. Moreover, the limitations mentioned doesn’t seem correct. For example, the study “DoÄźan Y, EkĹźi F, Karslıgil T, Bayram A. The Investigation of the Presence of Rotavirus and Adenovirus in Patients with Acute Gastroenteritis. Türk Mikrobiyol Cem Derg 2014,44,18- 22” reference #15 included inpatients/ hospitalized children with acute gastroenteritis but authors mentioned that exclusion of hospitalized children as a limitation. As this fundamentally affect the finding of this review work, authors need to clarify and address it properly.

Author Response

Response to Reviewer 1 Comments

Abstract

Needs extensive English language edition

Point1.For example, page 1 line 17&18 “Of the remaining 152 articles, only 39 18 studies were included in this meta-analysis.” is not clear. Without mentioning any data prior to this sentence, authors began “of the remaining…….”. It needs edition. On the other hand, one of the statements, “The effect size was estimated by reported with its 95% confidence interval.”, it would rather be paraphrased as “Effect size estimate was reported with 95% confidence interval.”. In General, there are many typographic, contextual, and grammatical errors which needs extensive language edition.

Response 1: The comprehensive English edition of the article was made by the MDPI English editing service (English Editing ID: English-22475). English grammar and word mistakes are corrected. The article screening was arranged to include all the steps from the beginning to the final state, and the flow diagram was rearranged.

The sentence (page 1 line 17&18)  has been changed as: We selected 38 out of 721 studies for our study.

The sentence (page 1 line20) has been changed as: Effect size estimate was reported with 95% confidence interval.

Introduction

Point 2:The statement “The 38 first infection tends to be the most violent because the body produces immunity against the virus.” Does not seem scientifically sound. Authors need to cite scientifically sound evidence that supports their statement or shall remove it.

Response 2: The first infection tends to be the most violent because the body produces immunity against the virus. Therefore, rotavirus infections are sporadic in adults [3,4].

Sentence removed

Methods

Point 3:In the inclusion criteria “The study period should include at least one year, the number of cases should be 100 and above, seasonal or month-by-case distribution, and the distribution by age groups.” Needs modification. For one thing, the authors included cross-sectional studies, therefore the study may not last for a complete one year. On the other hand, the number of cases indicated “100” is not true. Authors have included several studies, 7 studies to be specific, with less than 100 cases. If the authors intention is to specify about sample size, it has to be corrected accordingly. Authors need to revise also “Quality assessment” section as per the above comment related to the number of cases as quality assessment criteria. Meta-analysis section, the statement “The effect size [ES] was reported to be 95% confidence interval [CI]” needs modification.

Response 3:

The study period should include at least one year, the number of cases should be 100 and above, seasonal or month-by-case distribution, and the distribution by age groups.

Sentence removed

The sentence in the "Quality assessment " section has been revised to become the following.

 ‘’The quality of the studies documented in the articles was evaluated primarily based on the tests used to detect rotavirus, age groups, and seasonal distribution.’’

The sentence in the " Meta-analysis " section has been revised to become the following.

Effect size [ES]estimate was reported with 95% confidence interval [CI].

Results

Point 4:The flow chart has lots of flaws. Authors mentioned they had 152 study articles in first paragraph of the result section, but it is 158 in the flow diagram. Moreover, I do not really think they had only this few number of articles in their first database search before removal of duplication. Usually the database gives large number of articles but through a step wise screening the number of study articles included will be small. Moreover, the statements in the flow diagram comment boxes are misspelled. Generally, the flow diagram needs critical revision. Prevalence of rotavirus infection seems wrongly calculated. If we consider the overall prevalence in under five children to be 19%. There is no way that the prevalence in 0-2 YO =67% and 2-5= 25%. Because the overall prevalence would be >/= 25 according to the subgroup analysis data. Authors need to re-do the analysis. Authors shall put figures and tables in brackets when they cite in the result section.

Response 4:

The article screening was arranged to include all the steps from the beginning to the final state, and the flow diagram was rearranged.

All figures and tables are written in parentheses

Figure 3 and Figure 4 titles have been corrected.

Subgroup analyzes were re-performed and new forest plot charts were created.

Discussion

Point5:Authors have discussed their pooled rotavirus prevalence estimate with different reports and concluded that their prevalence is different from other study findings. For the differences authors mentioned “The reason for the different results from our study is the fiction and technique of the study” as a reason which is unclear and does not show how the study techniques were different.

Response:5

The reason for the different results from our study is the fiction and technique of the study.

 Sentence removed

Conclusion

Point 6:There are still spelling errors such as “route virus” to mean rotavirus. Authors need to check the whole document for language edition preferably by English language expert.

Response 6: English grammar and word mistakes are corrected

Limitations of the study

Point 7.This section shall precede the conclusion section. Moreover, the statements need rephrasing. Moreover, the limitations mentioned doesn’t seem correct. For example, the study “DoÄźan Y, EkĹźi F, Karslıgil T, Bayram A. The Investigation of the Presence of Rotavirus and Adenovirus in Patients with Acute Gastroenteritis. Türk Mikrobiyol Cem Derg 2014,44,18- 22” reference #15 included inpatients/ hospitalized children with acute gastroenteritis but authors mentioned that exclusion of hospitalized children as a limitation. As this fundamentally affects the finding of this review work, authors need to clarify and address it properly.

Response 7:This section has been placed before the conclusion section

The section has been revised to become the following;

Limitations of the study: This study limitation is that since the vaccination rates could not be reached in the articles, pre- and post-vaccination evaluation could not be made